# Lymphatic vessels interact dynamically with the hair follicle stem cell niche during skin regeneration *in vivo*

Daniel Peña-Jimenez[1,§], Silvia Fontenete[1,2,§], Diego Megias[3], Coral Fustero-Torre[4], Osvaldo Graña-Castro[4], Donatello Castellana[1,5], Robert Loewe[6] & Mirna Perez-Moreno[1,2,*] (iD)

## Abstract

**Lymphatic vessels are essential for skin fluid homeostasis and immune cell trafficking. Whether the lymphatic vasculature is associated with hair follicle regeneration is, however, unknown. Here, using steady and live imaging approaches in mouse skin, we show that lymphatic vessels distribute to the anterior permanent region of individual hair follicles, starting from development through all cycle stages and interconnecting neighboring follicles at the bulge level, in a stem cell-dependent manner. Lymphatic vessels further connect hair follicles in triads and dynamically flow across the skin. At the onset of the physiological stem cell activation, or upon pharmacological or genetic induction of hair follicle growth, lymphatic vessels transiently expand their caliber suggesting an increased tissue drainage capacity. Interestingly, the physiological caliber increase is associated with a distinct gene expression correlated with lymphatic vessel reorganization. Using mouse genetics, we show that lymphatic vessel depletion blocks hair follicle growth. Our findings point toward the lymphatic vasculature being important for hair follicle development, cycling, and organization, and define lymphatic vessels as stem cell niche components, coordinating connections at tissue-level, thus provide insight into their functional contribution to skin regeneration.**

**Keywords** hair cycle; hair follicle; lymphatic vessels; skin; stem cells
**Subject Categories** Development & Differentiation; Stem Cells & Regenerative Medicine; Vascular Biology & Angiogenesis
**The EMBO Journal (2019) 38: e101688**

See also: **CY Kam & V Greco** (October 2019)

## Introduction

Lymphatic vessels (LV) play fundamental homeostatic functions including the balanced transport of fluids and macromolecules, the local coordination of immune responses, and immune cell trafficking to regional lymph nodes (Skobe & Detmar, 2000). After years of scientific discovery, much has been learned about the distinctive characteristics of LV, including the molecular markers prospero-related homeobox 1 (Prox1), lymphatic vessel endothelial hyaluronan receptor 1 (LYVE1), and podoplanin, as well as critical regulatory signals that govern their development and fundamental functions in tissues (Wang & Oliver, 2010; Yang & Oliver, 2014; Zheng *et al*, 2014; Potente & Makinen, 2017).

In skin, LV are organized in structured polygonal patterns, consisting of one subcutaneous plexus, and a more superficial plexus located in the dermis, near the blood vessels (Braverman, 1989). Several studies have contributed to our understanding of the orderly organization of LV in the skin (Skobe & Detmar, 2000; Tripp *et al*, 2008). These studies have provided insight into the existence of branches of lymphatic capillaries that extend to the HF and drain into the subcutaneous collecting LV, presumably through connections of blind capillaries with the dermal papillae (dp) (Forbes, 1938), a condensate of dermal fibroblasts that provides a specialized microenvironment (Millar, 2002; Yang & Cotsarelis, 2010; Sennett & Rendl, 2012). LV may also facilitate the entry of immune cells to the HF epithelium, a source of chemokines that regulate the trafficking of epidermal Langerhans cells and dermal dendritic cells (Nagao *et al*, 2012), the distribution and differentiation of Langerhans cells (Wang *et al*, 2012), and the tropism of skin resident memory T cells (Adachi *et al*, 2015). However, despite the role of LV in facilitating immune cell trafficking to HF, less is known about their coordinated connections during HF cycling and functional implications.

---

1   Epithelial Cell Biology Group, Cancer Cell Biology Programme, Spanish Cancer Research Centre (CNIO), Madrid, Spain
2   Section of Cell Biology and Physiology, Department of Biology, University of Copenhagen, Copenhagen, Denmark
3   Confocal Microscopy Core Unit, Biotechnology Programme, Spanish Cancer Research Centre (CNIO), Madrid, Spain
4   Bioinformatics Unit, Structural Biology Programme, Spanish Cancer Research Centre (CNIO), Madrid, Spain
5   Center for Cooperative Research Biosciences (CIC bioGUNE), Derio Bizkaia, Spain
6   Department of Dermatology, Medical University of Vienna, Vienna, Austria
    *Corresponding author. Tel: +45 35 33 33 40; E-mail: mirna.pmoreno@bio.ku.dk
    §These authors contributed equally to this work
    [Correction added on 1 October 2019, after first online publication: the author affiliations have been updated.]

---

In adult skin, HF exhibits a lifetime polarized pattern of growth and regeneration across the tissue modulated by stimulatory and inhibitory signals (Plikus *et al*, 2011; Widelitz & Chuong, 2016). The cyclic regeneration of HF involves phases of growth (Anagen) via regression (Catagen) to relative quiescence (Telogen; Geyfman *et al*, 2015). The entry of resting HF into Anagen requires the activation of HFSC located in the HF bulge (Cotsarelis *et al*, 1990; Tumbar *et al*, 2004), and the expansion of their progenitors found in the secondary hair germ, giving rise to a new Anagen HF (Tumbar *et al*, 2004; Greco *et al*, 2009; Rompolas *et al*, 2012). Anagen HF grows until other instructive signals promote their regression giving rise to a new HF cycle. In past decades, a wealth of knowledge has yielded valuable insight into the role of major stimulatory and inhibitory signals in governing the orchestrated activation of the HF cycle (Blanpain & Fuchs, 2009; Lee & Tumbar, 2012; Plikus & Chuong, 2014), including local self-activation signals (Hsu *et al*, 2011), the contribution of other cells in the tissue macroenvironment (Brownell *et al*, 2011; Festa *et al*, 2011; Castellana *et al*, 2014; Rivera-Gonzalez *et al*, 2016; Ali *et al*, 2017), and long-range signaling waves across the skin (Plikus *et al*, 2011).

Blood vessels have also been found associated around HF exhibiting a coordinated dynamic reorganization during HF cycling (Mecklenburg *et al*, 2000; Yano *et al*, 2001). Also, HFSC closely associates with a venule annulus (Xiao *et al*, 2013). The occurrence of angiogenesis, the growth of new capillaries from pre-existing blood vessels, has been observed during Anagen (Mecklenburg *et al*, 2000). The epidermal expression of the vascular endothelial growth factor A (VEGF-A; Detmar, 1996) induces perifollicular angiogenesis and sustains HF growth; conversely, inhibition of VEGF-A leads to a delay in HF growth accompanied by reductions in HF size and perifollicular vascularization (Yano *et al*, 2001). Overall, these results exposed that HF and blood vessels form a functional operative system. In contrast, less is known about the role of LV in regulating this process. Here, we show that lymphatic capillaries are novel components of the HFSC niche, coordinating HF connections at tissue-level and provide insight into their functional association to the HF cycle.

# Results

## Lymphatic capillaries distribute in the vicinity of HFSC in a polarized manner

To investigate the association between LV and HFSC, we first defined the lymphatic distribution at HF in mouse back skin. To this end, we performed immunofluorescence analyses using antibodies to the lymphatic marker LYVE1 as well as to alpha smooth muscle actin (αSMA), enriched in the arrector pili muscle (apm). Lymphatic capillaries distributed in a polarized manner, aligned to the anterior side of the HF, opposite the distribution described for the apm (Fujiwara *et al*, 2011), ascending as blind capillaries along the HF permanent area toward the epidermis until the infundibulum (Fig 1A). This distribution was different from the one observed in the ear skin, which presented parallel lymphatic capillaries that were not associated with HF

(Fig EV1A). In the back skin, lymphatic capillaries densely distributed to the anterior side of HF at Tenascin-C areas (Fig 1B), a glycoprotein of the extracellular matrix (ECM) enriched in the HF bulge (Tumbar *et al*, 2004). To interrogate the existence of a lymphatic association with both embryo and adult HFSC, we next determined the lymphatic distance to Lhx2$^+$ SC (Fig 1C and D; Rhee *et al*, 2006) and CD34$^+$ HFSC (Fig 1E and F; Blanpain *et al*, 2004), respectively. Immunofluorescence analyses of back skin sections of P5 and P12 mice (Fig 1C and D) and P49 mice (Fig 1E and F) revealed that lymphatic capillaries distributed to the proximity of HFSC, within a distance inferior of 3 μm, within the ratio expected for components of the HFSC niche (Beck *et al*, 2011).

Next, we explored if functional HFSC creates a niche promoting the continuous association of LV with the HF bulge. Wnt signaling features prominently in HFSC and regulates their properties and functional activity (Choi *et al*, 2013). Thus, we reduced the expression of Wnt ligands in HFSC by ablating the expression of Wls in the Keratin 15 (K15) HFSC compartment, using the K15CrePR$^{+/T}$; Wls$^{Δ/Δ}$ conditional mouse model (Choi *et al*, 2013; Myung *et al*, 2013). Wntless (Wls) binds to Wnt ligands and controls their sorting and secretion (Carpenter *et al*, 2010). Under these conditions, HFSC remains quiescent, exhibiting a reduction in proliferation, but HF is largely maintained (Choi *et al*, 2013; Myung *et al*, 2013). Consistent with those findings, HF remained present in the skin (Fig 1G) despite the loss of Wls, as confirmed by RT–qPCR analyses of CD34$^+$ α6 integrin$^+$ FACS-isolated cells (Fig EV1B). LYVE1 immunofluorescence analyses revealed that the organized association of LV with HF was disrupted, and LV were found distributed parallel to the epidermis and distant from HF bulge areas (Fig 1G and H), reminiscent to the organization of LV in the ear skin (Fig EV1A). Overall, these results indicate that HFSC creates a niche for lymphatic endothelial cells, and sustain a polarized pattern of LV, which in turn, interconnect neighboring HF at the level of the HF bulge across the skin.

## Lymphatic capillaries start associating with HF during morphogenesis

To gain further insight into the establishment of the LV–HF association, we next turned to HF embryogenesis. HFSC emerges in the early HF placode stage at E15.5, giving rise to the hair germ (E16.5), hair pegs (E17.5), and embryo Anagen HF until the first postnatal HF cycle. Whole mount immunofluorescence analyses of LYVE1 at E15.5–E17.5 embryo stages (Fig 2A–C) revealed that from E15.5, when HF placodes were already visible, nascent networks of anastomosed LV start to form below areas of HF growth. HF starts to develop via epithelial–mesenchymal inductive signals stemming from the dermal papilla (dp) (Sennett & Rendl, 2012); however, no apparent lymphatic association with the dp was observed at E15.5 or E16.5, and LV were rather aligned in parallel to the epidermis, representative of lymphatic collecting vessels (Fig 2A and B). Interestingly, at E17.5 upward grows of lymphatic capillaries started to branch out from collecting vessels toward the papillary dermis and distributed at HF sites (Fig 2C). Overall, these results indicate that the development of HF is coupled with the recruitment of lymphatic capillaries to HF sites, likely subsequent to HF specification.

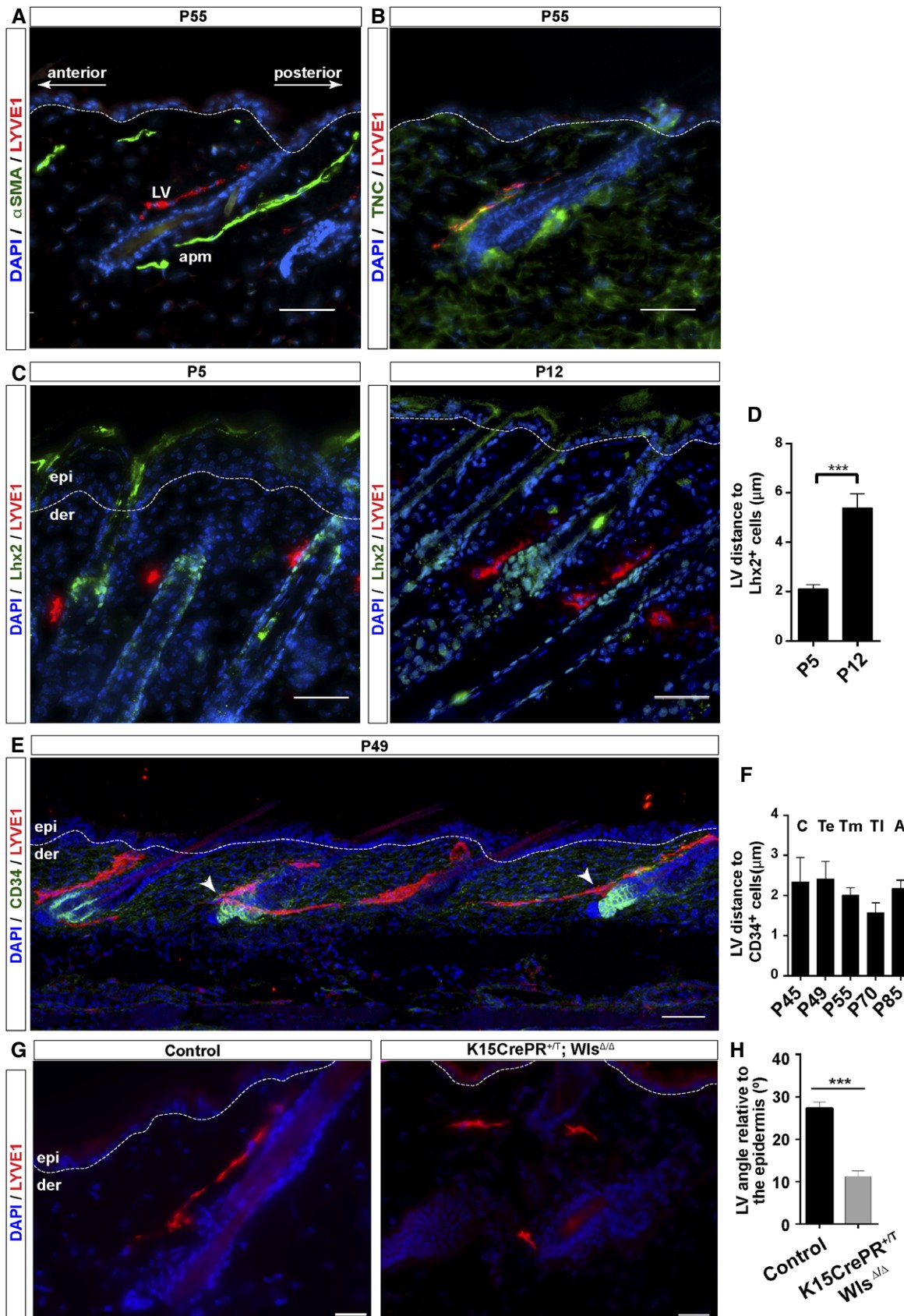

Figure 1.

◀

**Figure 1.  LV interact with HF in a functional HFSC niche-dependent manner.**

A   Adult mouse back skin sections immunostained for LYVE1 (red), αSMA (green), and counterstained with DAPI (blue). *n* = 3–4 skin samples per mouse, *n* = 3–4 mice. Scale bar, 50 μm. LV, lymphatic vessels; apm, arrector pili muscle.

B   Back skin sections of P55 mouse skin immunostained for LYVE1 (red), Tenascin-C (green), and counterstained with DAPI (blue). *n* = 3–4 skin samples per mouse, *n* = 3–4 mice. Scale bar, 50 μm. TNC, Tenascin-C.

C   Adult mouse back skin sections from different postnatal (P) days immunostained for LYVE1 (red), Lhx2 (green), and counterstained with DAPI (blue). *n* = 3–4 skin samples per mouse, n = 3–4 mice. Scale bar, 50 μm. epi, epidermis; der, dermis.

D   Histogram of the quantification of the LV distance to HFSC positive for Lhx2. *n* = 3–4 skin samples per mouse, *n* = 3–4 mice. Data represent the mean value ± SEM. ***P < 0.001 (Mann–Whitney *U*-test).

E   Back skin sections of P49 mouse skin immunostained for LYVE1 (red), CD34 (green), and counterstained with DAPI (blue). *n* = 3–4 skin samples per mouse, *n* = 3–4 mice. Scale bar, 100 μm. epi, epidermis; der, dermis. White arrowheads denote the proximity of lymphatic capillaries to HFSC.

F   Histogram of the quantification of the LV distance to HFSC positive for CD34. *n* = 3–4 skin samples per mouse, *n* = 3–4 mice. C, Catagen; Te, early Telogen; Tm, mid-Telogen; Tl, late Telogen; A, Anagen. Data represent the mean value ± SEM (Kruskal–Wallis test, Tukey's test).

G   Back skin sections of K15CrePR$^{+/T}$; Wls$^{\Delta/\Delta}$ and Control K15CrePR$^{+/+}$; Wls$^{flox/flox}$ mice treated with mifepristone from P7 during 12 weeks (P90), immunostained for LYVE1 (red), and counterstained with DAPI (blue). *n* = 3–4 skin samples per mouse, *n* = 3–4 mice. Scale bar, 100 μm. epi, epidermis; der, dermis.

H   Histogram of the quantification of the LV angle relative to the epidermis. *n* = 3–4 skin samples per mouse, *n* = 3–4 mice. Data represent the mean value ± SEM. ***P < 0.001 (Mann–Whitney *U*-test).

## Lymphatic vessels interconnect triads of HF across the back skin

To further analyze the organization of the lymphatic association with HF, we conducted whole mount immunofluorescence analyses of LYVE1 in adult skin sections (P70). This method allowed discerning an additional arrangement level, where individual LV–HF units associated further into triads (Fig 3A and Movie EV1). The patterned polygonal organization of lymphatic-HF domains across the skin was more evident in 3D projection planes (Fig 3B and Movie EV2). Lymphatic capillaries were found associated along the permanent portion of individual HF. At the level of the HF bulge, lymphatic capillaries radiated and converged interconnecting three HF units. These units presented a common extending lymphatic capillary from each HF triad (Fig 3B and Movie EV2). The mechanisms involved in the lymphatic-HF patterning are interesting questions for the future, but globally, these results uncharted the existence of coordinated lymphatic arrays surrounding and interconnecting triads of HF across the skin.

## Lymphatic vessels continuously associate with the HF bulge during the HF cycle

We next explored whether LV change their distribution to the HF bulge areas during the postnatal HF cycle. These analyses were performed in isolated back skin sections from matched skin areas at defined stages of the HF cycle, including the postnatal HF morphogenesis (postnatal days 5–16, P5–P16), and the first (P23–P45) and the second (P49–P85) HF cycle (Fig EV2; Muller-Rover *et al*, 2001). The latter exhibits a more extended Telogen that lasts for 3–4 weeks; therefore, to perform our comparative analyses we subdivided the second Telogen into early Telogen (Te, P49), mid-Telogen (Tm, P55), late Telogen (Tl, P69), and included an Anagen stage (A$_{VI}$, P85; Muller-Rover *et al*, 2001). These phases corresponded to the refractory and competent Telogen phases, as previously documented (Plikus *et al*, 2008).

Interestingly, at all HF stages, LV remained distributed in a polarized manner, positioned at the anterior side of the permanent region of the HF opposite to the apm, interconnecting neighboring HF at the level of the HF bulge (Fig EV2A). The greater lymphatic density was localized at the HF permanent region, exhibiting a significantly

higher area at postnatal HF morphogenesis stages (P5–P16; Fig EV2B). Consistent with this distribution, the relative LV/HF length increased from late Anagen stages to Telogen, when HF mostly consists of the permanent region (Fig EV2C). Conversely, this relative length was reduced during the transition from Telogen to Anagen (Fig EV2C). Except for the HF stages P5, P12, and P16, no changes in LV proliferation were observed (Fig EV2D), suggesting that LV were still growing and reorganizing to developing HF. Also, no changes in LV cell death were observed during the HF cycle (Fig EV2E).

These studies exposed that LV establish a continuous connection between neighboring HF at the level of the HF bulge across the skin throughout all phases of the HF cycle.

## Lymphatic vessels dynamically flow across triads of HF in the back skin

To further pursue the existence of dynamic lymphatic flow between neighboring HF triads across the skin, we analyzed the lymphatic network by intravital microscopy, using a Prox1-CreERT2; ROSA26-LSL-eYFP reporter mouse. This transgenic mouse line expresses the tamoxifen-inducible Cre recombinase (CreER) under the control of the Prox1 gene promoter (Bazigou *et al*, 2011), under a Rosa26-LSL-eYPP background. Prox1 is essential for LV development, and it is expressed throughout life, providing lymphatic identity (Wigle & Oliver, 1999; Hong *et al*, 2002). These powerful tools allowed us to explore the dynamic association of lymphatic capillary networks with HF in living skin tissue. Our intravital microscopy analyses fully evidenced the continuum-patterned organization of lymphatic networks around HF in the back skin (Fig 3C). These analyses further allowed the visualization of triads of HF interconnected by lymphatic capillaries, aligned in parallel rows with an anterior to posterior disposition. Strikingly, the HF triads in each row interconnected with neighboring parallel rows, mainly through a LV stemming from one HF triad unit (Fig 3C), consistent with our prior observations (Fig 3B and Movie EV2).

We next assessed whether LV dynamically streamed into the continuum-patterned lymphatic networks around HF in the back skin. To this end, TRITC–Dextran was administered in the mouse back skin followed by intravital microscopy. These results allowed

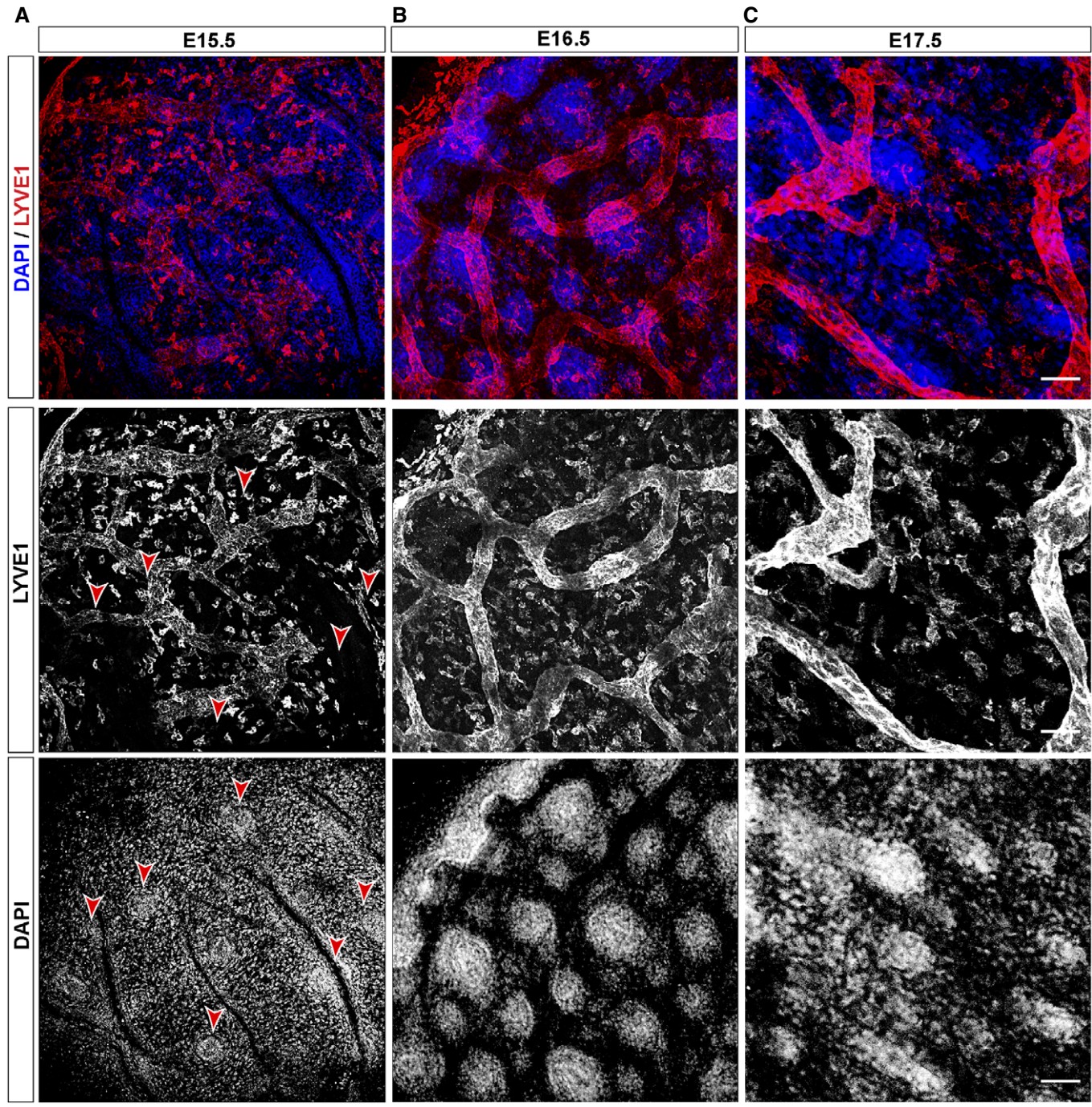

**Figure 2. LV associate with HF during development.**

A–C Distribution of LV and HF at the mouse embryo stage E15.5 (A), E16.5 (B), and E17.5 (C). Maximum projection images of whole mount immunofluorescence analyses using LYVE1 (red) as lymphatic endothelial marker and counterstained with DAPI (blue). Red arrowheads denote HF placodes. $n$ = 3–4 embryos. Scale bar, 50 μm.

the visualization of a continuous lymphatic flow interconnecting adjacent HF rows (Movie EV3).

These results exposed the existence of a dynamic HF communication through lymphatic vascularization, which potentially facilitates the spreading of signaling waves and immune cell trafficking across HF in the back skin.

**Lymphatic endothelial cells transiently increase their caliber at the onset of HF stem cell activation**

Our previous findings raised the possibility that LV undergo dynamic flow changes at different stages of the HF cycle. We investigated this aspect by measuring the LV caliber in back skin sections during the

first and the second HF cycle, conducting LYVE1 immunofluorescence analyses. Intriguingly, the LV caliber was more pronounced at the onset of the Telogen to Anagen transition (Fig 4A and B), without exhibiting signs of proliferation or cell death (Fig EV2D and E). We next inspected more closely the LV morphology and observed that at late stages of Telogen, LV appeared more fenestrated displaying membrane protrusions compared to the more continuous and tight capillaries observed during Anagen. This structural variation, exhibiting a wide and irregular lumen, presumably accounts for regional differences in capillary permeability (O'Driscoll, 1992) influencing vascular exchange and increase tissue drainage capacity (Aebischer et al, 2014; Betterman & Harvey, 2016).

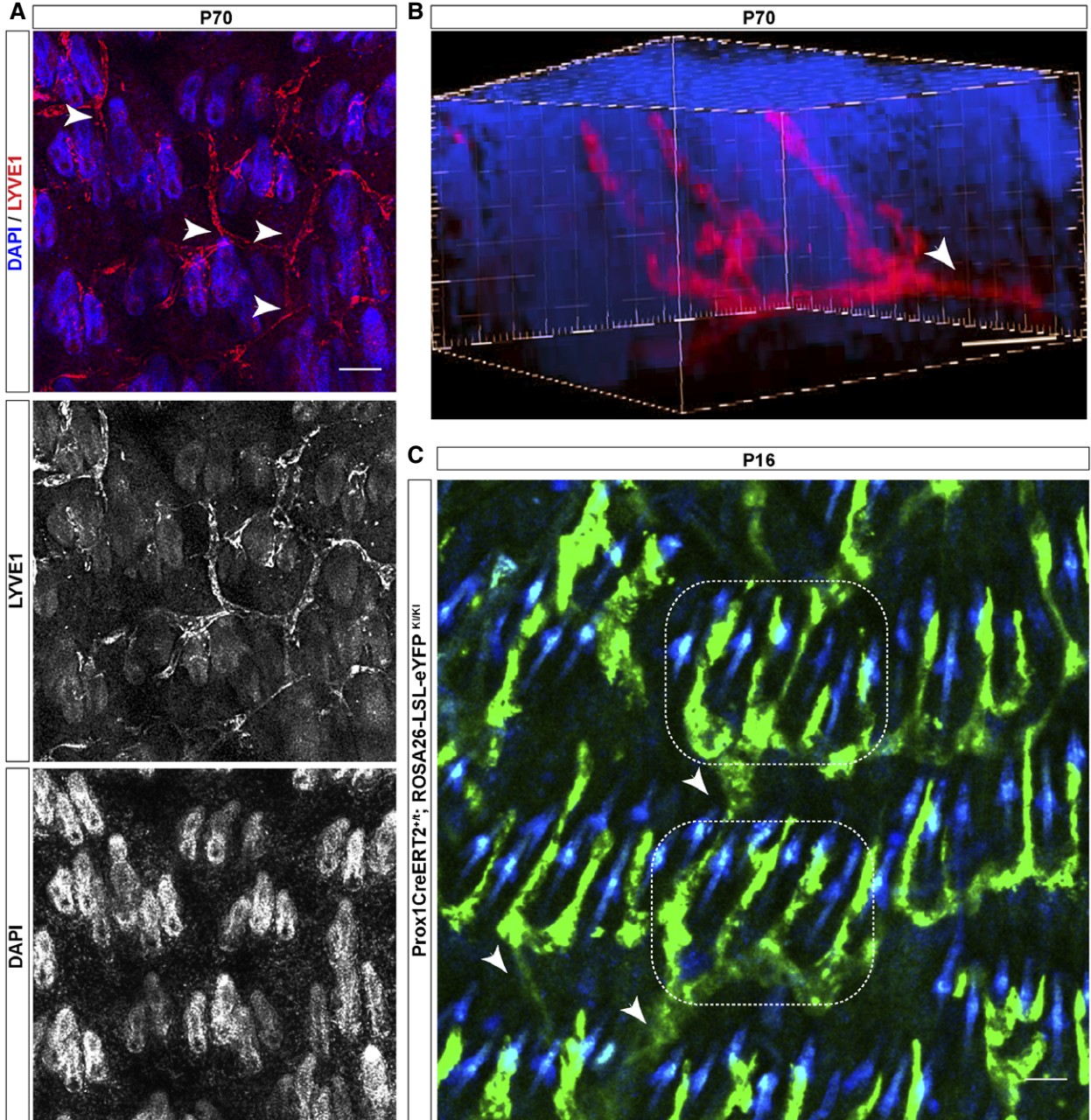

**Figure 3.  LV association with HF during the postnatal HF cycle.**

A    LV associated with individual HF further organize associating HF triads, which connect with other HF triads across the skin. P70, 70-day-old mice. n = 3–4 mice. White arrowheads denote capillaries stemming from HF triads. Scale bar, 100 μm.

B    3D reconstructions of whole skin mounts showing a triad of HF connected by LV. n = 3–4 mice. Scale bar, 50 μm. White arrowhead denotes an extending lymphatic capillary from a HF triad.

C    Images of intravital microscopy analyses showing aligned HF rows in the back skin interconnected by LV in the Prox1CreERT2; Rosa-LSL-eYPF mice. Dotted boxes denote HF triads; white arrowheads denote capillaries stemming from HF triads interconnecting to other HF triads in adjacent rows. n = 3–4 mice. Scale bar, 50 μm.

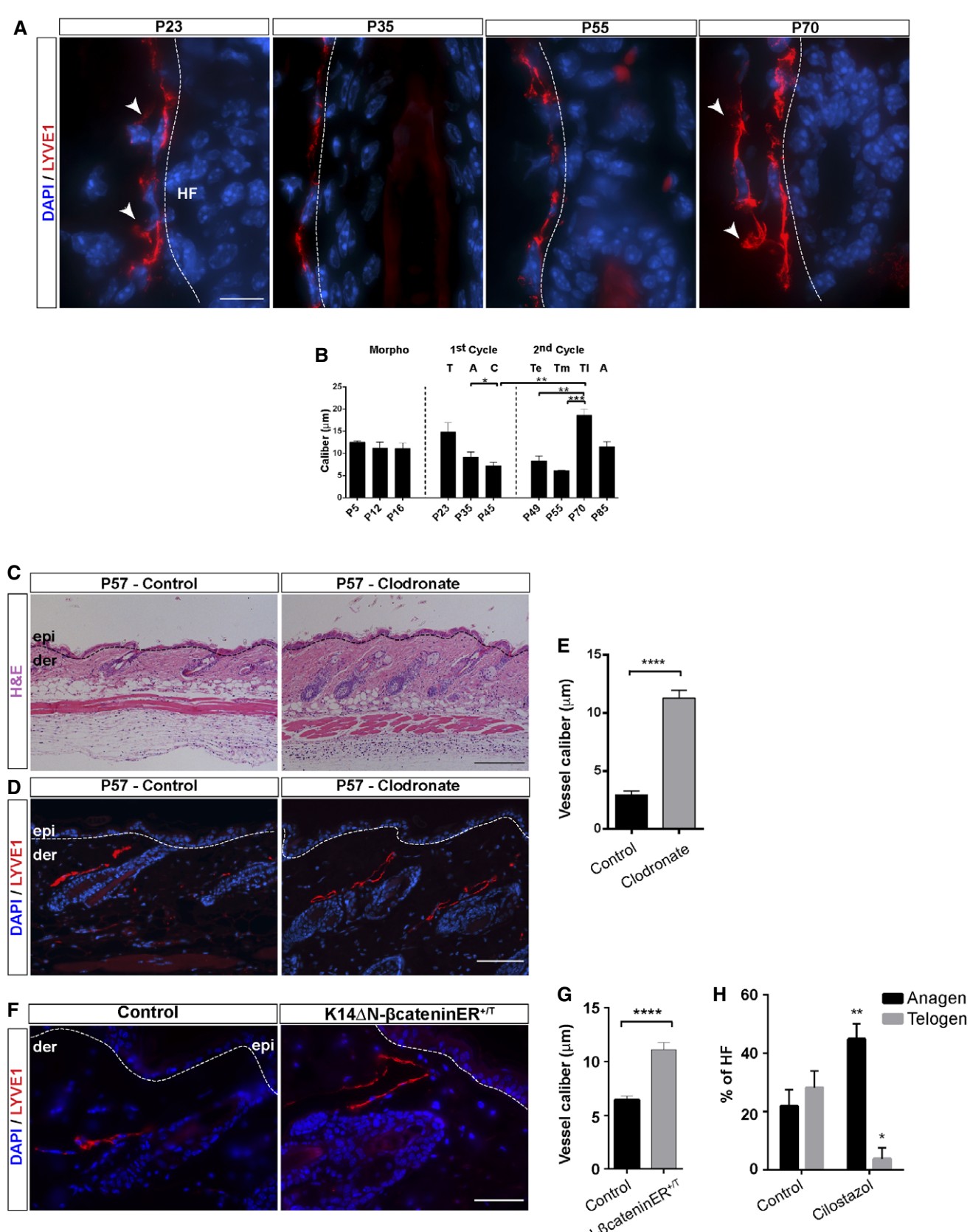

Figure 4.

**Figure 4. Dynamic reorganization of LV during the HF cycle.**

- A    Adult back skin sections from different postnatal (P) days immunostained for LYVE1 (red) and counterstained with DAPI (blue). $n = 3-4$ skin samples per mouse, $n = 3-4$ mice. Scale bar, 10 μm. White arrowheads denote LV membrane protrusion and fenestrated areas. LV, lymphatic vessels; HF, hair follicle.
- B    Histogram of the caliber (μm) of LV at different postnatal days. $n = 3-4$ skin samples per mouse, $n = 3-4$ mice. A, Anagen; C, Catagen; T, Telogen; Te, early Telogen; Tm, mid-Telogen; Tl, late Telogen. The data shown represent the mean value ± SEM. *$P < 0.05$; **$P < 0.01$; ***$P < 0.001$ (one-way ANOVA, Tukey's test).
- C    H&E staining of back skin sections from mice treated intradermally with Control or Clodronate liposomes. $n = 3-4$ skin samples per mouse, $n = 3-4$ mice. Scale bar, 200 μm. epi, epidermis; der, dermis.
- D, E    LYVE1 immunofluorescence (red) counterstained with DAPI (blue) (D) and histogram of the caliber (μm) of LV (E) in back skin sections from mice treated intradermally with Control or Clodronate liposomes. $n = 3-4$ skin samples per mouse, $n = 3-4$ mice. Scale bar, 100 μm. epi, epidermis; der, dermis. The data shown represent the mean value ± SEM. ****$P < 0.0001$ (Mann–Whitney $U$-test).
- F    Back skin sections of Controls and K14ΔNβ-cateninER$^{+/T}$ mice immunostained for LYVE1 (red) and counterstained with DAPI (blue). $n = 3-4$ skin samples per mouse, $n = 3-4$ mice. Scale bar, 50 μm. epi, epidermis; der, dermis.
- G    Histogram of the LV caliber in skin sections from Controls and K14ΔNβ-cateninER$^{+/T}$ mice. $n = 3-4$ skin samples per mouse, $n = 3-4$ mice. Data represent the mean value ± SEM. ****$P < 0.0001$ (Mann–Whitney $U$-test).
- H    Histogram of the percentage of HF in Telogen and Anagen present in the skin of control mice or mice treated with Cilostazol. $n = 3-4$ skin samples per mouse, $n = 3-4$ mice. Data represent the mean value ± SEM. *$P < 0.05$; **$P < 0.01$ (Mann–Whitney $U$-test).

We next explored whether the induction of HF growth induces the transitory increase in the caliber of LV. To this end, to avoid severing LV and the induction of an inflammatory response, we did not conduct hair plucking experiments to synchronize HF, but rather stimulated the Telogen to Anagen transition by reducing the number of perifollicular macrophages at early Telogen with Clodronate liposomes (Fig 4C–E), as previously described (Castellana et al, 2014). Under these conditions, LV surrounding the precocious Anagen HF displayed an increase in their caliber compared to controls (Fig 4D and E). We further analyzed the connection between HFSC cell proliferation and the expansion of LV caliber in K14Cre$^{+/T}$, ΔNβ-catenin$^{lox/lox}$ mouse skin samples. This model is characterized by continuous HFSC proliferation and pilomatricoma formation (Lowry et al, 2005; Jensen et al, 2009). Immunofluorescence analyses for LYVE1 revealed a significant increase in LV caliber under conditions of continuous HF growth (Fig 4F and G). To interrogate whether a sustained increased in vascular flow prompts the entry of Telogen HF into Anagen, we pharmacologically treated mice in early Telogen (P49) with Cilostazol (Kimura et al, 2014); as previously documented (Choi et al, 2018), increasing the vascular flow led to precocious HF growth and an increase in the percentage of HF in Anagen (Fig 4H). Overall, these results exposed the existence of networks of LV that dynamically reorganize increasing their caliber upon activation of HFSC.

## Transcriptome analysis of lymphatic endothelial cells at the onset of HFSC activation

Our prior results prompted us to analyze the existence of a differential gene expression in LV during the physiological HF Telogen to Anagen transition, when LV exhibit an expanded caliber (Fig 4A and B). We focused on the physiological Tm (P55) and Tl (P70) phases of the second HF cycle as HF progress to Anagen. To this end, we FACS-isolated eYFP$^+$ dermal LV from the back skin of Prox1-CreERT2$^{+/T}$; Rosa26-LSL-eYFP$^{KI/KI}$ mice, where 60% of the cells represent lymphatic endothelial cells (Bianchi et al, 2015). We avoided including the dermal subcutaneous collector LV, through mechanical removal before tissue digestion. To comprehensively define changes in the molecular traits of isolated cells, we conducted RNA sequencing (RNA-seq) analyses. The transcriptome profiles revealed the expression of genes differentially regulated between the analyzed populations. We first determined the existence of differentially expressed pathways in Tl compared with Tm phases,

establishing LV gene expression rankings using gene set enrichment analyses (GSEA) compared with public annotations from Reactome, BioCarta, Kyoto Encyclopedia of Genes and Genomes (KEGG), and Gene Ontology (GO) databases. The majority of the differentially regulated genes (FDR $q$-values < 0.25) were related to cell adhesion, cytoskeleton, and lymphatic organization/axon guidance (Fig 5A). Further analyses revealed that 310 genes were upregulated and 562 downregulated with a 2.8-fold change ($\log_2$ fold change > 1.5 or < −1.5) at Tl (P70) compared with Tm (P55). Enrichr analyses (Chen et al, 2013; Kuleshov et al, 2016) were then used to classify the major represented categories, which exposed the differential expression of genes related to LV remodeling (Fig 5B and C), in agreement with the transitory morphological changes observed in LV at the onset of HFSC activation. The genes encoded proteins involved in ECM, cytoskeleton and adhesion processes, lymphatic remodeling and axon guidance as well as genes involved in intracellular signaling processes. Some transcripts of the most over-represented pathways were validated by double in situ hybridization analyses for LYVE1 and candidate transcripts (Figs 5D–G and EV3A–D), or by double immunofluorescence analyses (Figs 5H and I, and EV3E and F). Transcripts encoding the ECM adhesion protein integrin α5 (ITG5A) were found upregulated (Figs 5D and EV3A) at Tl (P70) compared with Tm (P55), while the ECM secreted proteoglycan decorin (DCN) levels tended to be downregulated although non-significant (Figs 5E and EV3B). Also, an increased immunofluorescence intensity for the adhesion protein Jup was observed in the expanded and irregular LV lumen surrounding HF at P70 compared with P55 (Figs 5H and EV3E). The expression of transcripts encoding for the lymphatic remodeling and axon guidance proteins Polycystin-1 (PKD1; Figs 5F and EV3C), Plexin D1 (PLXND1; Figs 5G and EV3D), and the immunofluorescence intensity of Emilin1 (Figs 5I and EV3F) was also found upregulated at Tl (P70) compared with Tm (P55).

These genes have been involved in the regulation of lymphatic remodeling, lumen organization, and permeability, and taken together, these results support the existence of a transitory dynamic reorganization of LV at the onset of HFSC activation.

## Depletion of LV blocks both the pharmacological induction and the physiological HF growth

To get insight into the functional association between LV and the HF cycle, we pharmacologically stimulated HF growth, followed by

loss of function approaches using the mouse genetic model Prox1-CreERT2$^{+/T}$;Rosa26-LSL-iDTR$^{KI/KI}$ mice to conditionally ablate LV. Prox1CreERT2$^{+/T}$;Rosa26-LSL-iDTR$^{KI/KI}$ and controls were first

treated at early Telogen (P49) with cyclosporine A (CSA), a potent hair growth stimulator (Paus *et al*, 1989; Gafter-Gvili *et al*, 2003; Horsley *et al*, 2008), which activates quiescent HFSC through the

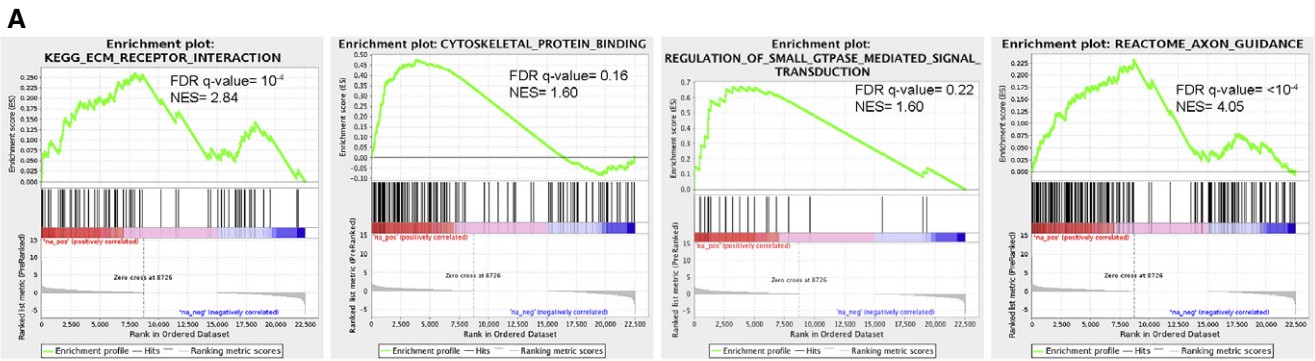

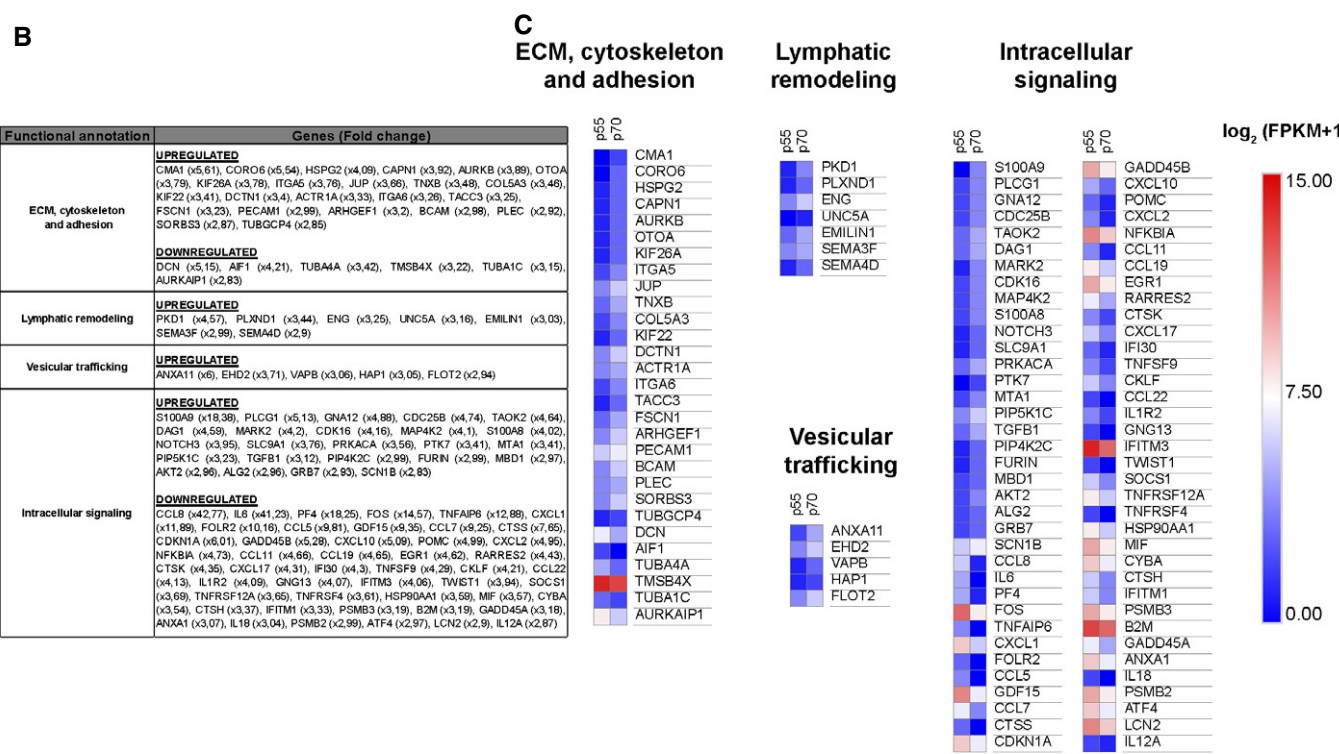

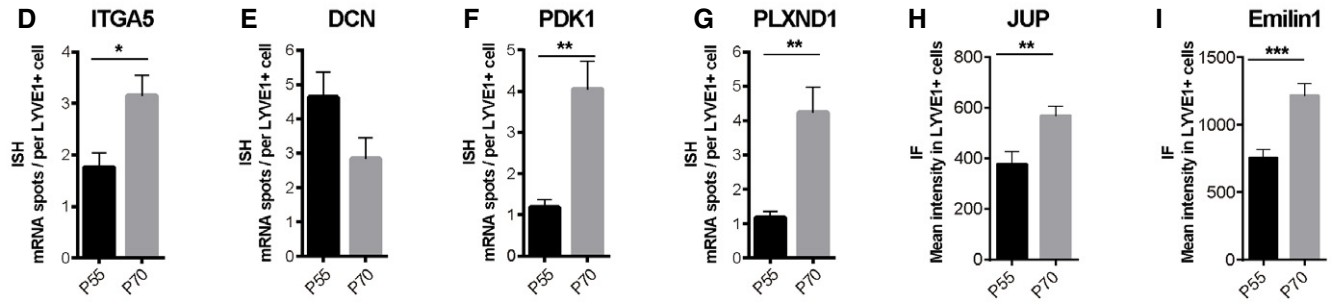

**Figure 5.**

**Figure 5. Transcriptome analysis of LV in late Telogen compared with mid-Telogen.**

A    Gene set enrichment plots (GSEA) of LV using public annotations showing the enrichment of selected gene signatures involved in different pathways in late Telogen (P69, Tl) compared with mid-Telogen (P55, Tm). NES: normalized enrichment score. The false discovery rate (FDR; $q$-value) is indicated.

B    Table of differentially expressed genes (DEGs), in LV during late Telogen compared to mid-Telogen, relative to biological processes, analyzed using the Enrichr platform (Chen *et al*, 2013; Kuleshov *et al*, 2016). Upregulated (red) and downregulated (blue). Data shown represent the fold change of the genes.

C    Heatmap diagram of DEGs in LV during late Telogen compared with mid-Telogen, relative to biological processes, analyzed using the Enrichr platform (Chen *et al*, 2013; Kuleshov *et al*, 2016). Data shown represent the log$_2$ (FPKM + 1).

D–G  Histograms of RNA ISH analyses by RNAscope to quantify the level of expression of *ITGA5* (D), *DCN* (E), *PKD1* (F), and *PLXND1* (G), assessed by the number of mRNA spots present in LYVE1$^+$ cells in the vicinity of HF in P55 and P70 mouse back skin. $n$ = average of 30 LYVE1$^+$ cells/sample, $n$ = 3 mice. Data represent the mean value ± SEM. *$P$ < 0.05; **$P$ < 0.01 (D, F, and G, unpaired Student's $t$-test; E, Mann–Whitney $U$-test).

H, I    Histograms represent the mean fluorescence intensity for Jup (H) and Emilin1 (I) in LYVE1$^+$ cells in the vicinity of HF in P55 and P70 mouse back skin. $n$ = 3–4 skin samples per mouse, $n$ = 3–4 mice. Data represent the mean value ± SEM. **$P$ < 0.01, ***$P$ < 0.001 (H, Mann–Whitney $U$-test; I, unpaired Student's $t$-test).

inhibition of calcineurin and the transcription factor of activated T cells c1 (NFATc1; Gafter-Gvili *et al*, 2003; Horsley *et al*, 2008). Consistent with those prior findings, the precocious growth of HF in CSA-treated mice was fully distinctive at the end of the experiment compared to controls, as observed by histological approaches (Fig 6A and B). Precocious HF exhibited a significant increase in proliferation, as evidenced by immunohistological analyses of the proliferation marker Ki67 (Fig 6C and D). Moreover, in agreement with our prior findings (Fig 4), the induction of precocious HF growth promoted a significant increase in LV caliber (Fig 6E and F).

CSA-treated Prox1CreERT2-iDTR mice were further treated with intradermal injections of diphtheria toxin (DT) for 3 days before the end of the CSA treatment. The DT treatment leads to a significant ablation of LV, as confirmed by reductions in lymphatic drainage, supported by the impairment of Evans Blue dye clearance in skin (Fig EV4A) and LYVE1 immunofluorescence analyses (Fig EV4B). Strikingly, the loss of LV precluded the precocious activation of HF growth induced by CSA, as evidenced by histological features (Fig 6A and B) and a significant reduction of Ki67$^+$ proliferating cells (Fig 6C and D), in comparison with CSA-treated controls. As an additional control, we also included Prox1CreERT2$^{+/T}$; Rosa26-LSL-iDTR$^{KI/KI}$ mice at early Telogen that were not treated with CSA, and no changes in HF organization of growth were observed upon LV ablation with DT (Fig EV5A and B). To further analyze the implications of LV in supporting HF growth, we ablated LV with DT for 3 days during the physiological Anagen of the first HF cycle (Fig 6G–J). The results show that the ablation of LV (Fig EV5C) prompted to a collapse of growing HF (Fig 6G and H), accompanied with cell death (Fig 6I and J) and loss of the growing HF differentiated layers (Fig EV5C–F).

These results exposed a functional connection between LV and HF growth.

## Discussion

Mammalian HF undergoes a lifetime cyclic regeneration, and several discoveries have increased our understanding on the regulation of this significant model of organ regeneration, both at individual HF level, through coordinated interactions with other cells at the HFSC niche or molecular signaling across adjacent HF across the tissue (Widelitz & Chuong, 2016; Gonzales & Fuchs, 2017; Guasch, 2017). Here, we now show that within a single HF, HFSC associates with LV capillaries, starting from developmental HF stages. We further

uncovered the existence of a dynamic patterned association of LV with adjacent HF across the skin and defined that the loss of LV abrogates the pharmacological induction of HF regeneration. These findings position LV as new elements of the HFSC niche and open new interesting questions about the potential roles of LV in regulating individual and adjacent HF at different stages of the physiological HF cycle.

We first discovered that the interaction of LV with individual HF is polarized and occurs along the anterior permanent HF region (Fig 1). Of note, we did not observe a preferential association of LV with particular HF types, as it has been observed for mechanosensory neurons (Li *et al*, 2011). It remains to be investigated whether the polarized LV–HF association is dependent on the existence of differentially expressed molecules at the anterior/posterior sides of the HF, thereby creating a distinct connection similar to the one documented between HF and the apm (Fujiwara *et al*, 2011). The nature of the lymphatic endothelial cells associated with HF is also intriguing, since they are plastic and heterogeneous among organs (Jang *et al*, 2013; Martinez-Corral *et al*, 2015; Petrova & Koh, 2018). Thus, a defined population may exert specific functions at HF, compared to lymphatic endothelial cells distributing at other anatomical locations within the skin.

It is generally well-acknowledged that HFSC creates a special niche microenvironment, where the interactions with other cells within the niche are important for HFSC organization and function (Gonzales & Fuchs, 2017; Guasch, 2017). In this regard, we observed that a functional HFSC niche is required to sustain the polarized LV–HF association in the back skin, through the expression of Wnt ligands. In the absence of HFSC-derived Wnts, LV dissociate from HF allocating parallel to the epidermis (Fig 1G and H), in a similar disposition to the one observed in the ear skin (Fig EV1A), supporting the existence of unique LV regional microenvironments. Interestingly, the lymphatic capillaries that associate with individual HF further branch at the level of HFSC, converging into a single LV for every three HF units, which in turn interconnect with other HF triads across the skin (Fig 3). This patterned LV organization resembles the one documented for dermal blood vessels distributed at the dermal plexus (Sada *et al*, 2016). At this location, blood vessels set different regional areas and contribute to the maintenance of two independent epidermal stem cell populations. However, respecting HFSC, blood vessels are distributed in a distinct manner, forming a venule annulus starting from morphogenesis (Xiao *et al*, 2013). It remains to be explored whether LV exert a defined regulatory role on a particular SC population, since distinct populations of SC exhibiting different potential arise from morphogenesis to later

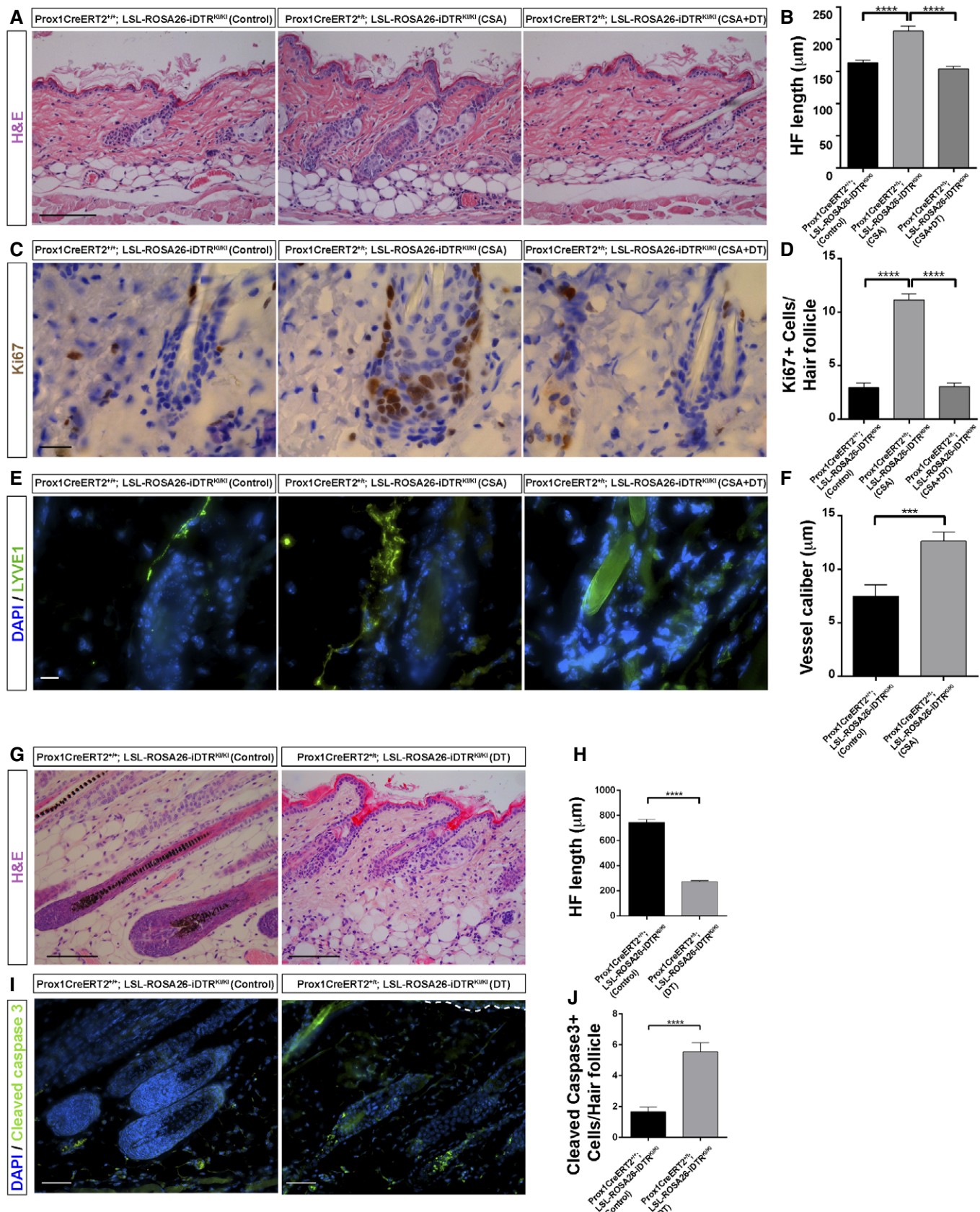

**Figure 6.**

◀

**Figure 6. Depletion of LV blocks both the pharmacological induction and the physiological HF growth.**

A, B H&E staining (A) and histogram of the HF length (B) in adult back skin sections from Prox1CreERT2[+/+]; LSL-ROSA26-iDTR[KI/KI] mice injected with vehicles (Control), Prox1CreERT2[+/T]; LSL-ROSA26-iDTR[KI/KI] mice treated with CSA and vehicle (CSA); and Prox1CreERT2[+/+]; LSL-ROSA26-iDTR[KI/KI] treated with tamoxifen, CSA, and intradermal diphtheria toxin (CSA + DT) starting from early Telogen (P49) and analyzed at the end of the treatments (P58). $n$ = 3–4 skin samples per mouse, $n$ = 3–4 mice. Scale bar, 200 μm. Data represent the mean value ± SEM. ****$P$ < 0.0001 (one-way ANOVA, Tukey's test).

C, D Ki67 immunostaining (C) and histogram of the number of Ki67[+] cells per HF (D) in adult back skin sections from Prox1CreERT2[+/+]; LSL-ROSA26-iDTR[KI/KI] mice injected with vehicles (Control), Prox1CreERT2[+/T]; LSL-ROSA26-iDTR[KI/KI] mice treated with CSA and vehicle; and Prox1CreERT2[+/+]; LSL-ROSA26-iDTR[KI/KI] treated with tamoxifen, CSA, and intradermal DT, starting from early Telogen (P49) and analyzed at the end of the treatments (P58). $n$ = 3–4 skin samples per mouse, $n$ = 3–4 mice. Scale bar, 20 μm. Data represent the mean value ± SEM. ****$P$ < 0.0001 (one-way ANOVA, Tukey's test).

E, F LYVE1 immunostaining (E) and histogram of the LV caliber (F) analyzed in adult back skin sections from Prox1CreERT2[+/+]; LSL-ROSA26-iDTR[KI/KI] mice injected with vehicles (Control), Prox1CreERT2[+/T]; LSL-ROSA26-iDTR[KI/KI] mice treated with CSA and vehicle (CSA); and Prox1CreERT2[+/+]; LSL-ROSA26-iDTR[KI/KI] mice treated with tamoxifen, CSA, and intradermal DT, starting from early Telogen (P49) and analyzed at the end of the treatments (P58). Scale bar, 10 μm. $n$ = 3–4 skin samples per mouse, $n$ = 3–4 mice. Data represent the mean value ± SEM. ***$P$ < 0.001 (Mann–Whitney $U$-test).

G, H H&E staining (G) and histogram of the HF length (H) in adult back skin sections from Prox1CreERT2[+/+]; LSL-ROSA26-iDTR[KI/KI] mice (Control) and Prox1CreERT2[+/T]; LSL-ROSA26-iDTR[KI/KI] mice treated with tamoxifen and intradermal DT, starting from Anagen (P30) and analyzed at the end of the treatments (P37). $n$ = 3–4 skin samples per mouse, $n$ = 3–4 mice. Scale bar, 200 μm. Data represent the mean value ± SEM. ****$P$ < 0.0001 (unpaired Student's $t$-test).

I, J Cleaved caspase-3 immunostaining (I) and histogram of the number of cleaved caspase-3[+] cells per HF (J) in adult back skin sections from Prox1CreERT2[+/+]; LSL-ROSA26-iDTR[KI/KI] mice (Control) and Prox1CreERT2[+/T]; LSL-ROSA26-iDTR[KI/KI] mice treated with tamoxifen and intradermal DT, starting from Anagen (P30) and analyzed at the end of the treatments (P37). Scale bar, 50 μm. $n$ = 3–4 skin samples per mouse, $n$ = 3–4 mice. Data represent the mean value ± SEM. ****$P$ < 0.0001 (Mann–Whitney $U$-test).

localize along the HF permanent region and at the HF bulge (Guasch, 2017). The LV connection with developing HF from embryo stages likely occurs after HF specification (Fig 2). This association could potentially implicate that LV regulate specific populations of SC after their specification favoring the subsequent HF development by regulatory signals, or through facilitating the access to nutrients or immune cell trafficking to the developing HF.

In adult skin, we observed that LV dynamically flow through neighboring HF across the skin. Interestingly, the LV caliber transiently expanded at the onset of HFSC activation (Fig 4), suggesting an increased tissue drainage capacity. Studies combining live imaging approaches and biophysical analyses of LV will be instrumental for defining spatiotemporal changes in LV dynamics at different stages of the HF cycle. The dynamic flow of LV through HF likely facilitates the distribution of molecules and immune cells across the skin. Although it remains unclear whether changes in the diffusion of soluble or cellular components through LV is linked to HF cycle regulation, our results now add LV as new roads connecting HF across the skin and pave the way for further investigation. In this regard, it is well acknowledged that the coordinated propagation of stimulatory/inhibitory signals across the skin regulates the HF cycle (Widelitz & Chuong, 2016). Interestingly, measurements of the decline levels of diffusible molecules, including growth factors, morphogens, and cytokines, predicted that their range of action is restricted to no more than 100 μm (Teleman & Cohen, 2000; Weber *et al*, 2013). However, in the back skin it has been observed that the effect of diffusible molecules is exerted at more than 1 mm further from their initial expression site (Chen *et al*, 2015). Thus, LV could potentially partake in this event aiding the diffusion of molecules at longer distances. Also, LV functional roles in draining tissue fluids could further modulate the concentration level of molecular inhibitors or activators at specific phases of the HF cycle.

LV may also regulate immune cell trafficking to HF areas, although immune privilege areas are normally devoid of LV (Paus *et al*, 2003). Indeed, it has been recently reported that Foxp3[+] T regulatory cells, which traffic through LV (Hunter *et al*, 2016), localize to HF, and regulate HFSC proliferation (Ali *et al*, 2017). Perifollicular macrophages have also been shown to contribute to the activation of HFSC through the expression of Wnt ligands

(Castellana *et al*, 2014), and myeloid Wnt ligands regulate the development of LV (Muley *et al*, 2017). Moreover, macrophages regulate LV caliber (Gordon *et al*, 2010), through the expression of Wnt ligands (Muley *et al*, 2017). Interestingly, our observations on the transient increment of the LV caliber at the onset of HFSC activation also coincide with the documented expression of macrophage-derived Wnt ligands and the activation of HFSC (Castellana *et al*, 2014). Thus, changes in the LV caliber could exert regulatory functions through the spatiotemporal diffusion of morphogens and the influx and outflow of immune cells to HF at specific phases of the HF cycle.

Our results exposed a marked increase in the LV caliber during the physiological late Telogen to Anagen transition (Fig 4). The induction of HFSC activation by modulating macrophage levels or upon pharmacological induction with CSA (Figs 4 and 6) exposed a functional connection between HFSC and LV caliber expansion upon HFSC activation. These changes were sustained under conditions of continuous HF growth as observed in the skin of K14Cre[+/T], ΔNβ-catenin[lox/lox] mice (Fig 4). During the physiological Telogen to Anagen transition, these LV changes were associated with a more irregular and fenestrated lumen morphology, in agreement with the distinct expression of genes encoding proteins mainly involved in lymphatic cytoskeletal remodeling and matrix cell adhesion (Fig 5A and B), suggesting the occurrence of mechanotransduction processes, changes in interstitial pressure, and modulation of tissue drainage (Planas-Paz & Lammert, 2013; Sabine *et al*, 2016). Whether these modifications could also lead to changes in the HFSC niche stiffness, which are known to regulate HFSC behavior (Lane *et al*, 2014), needs further investigation. The physiological roles of the identified LV signatures, as well as their potential roles in the activation/inhibition of HF regeneration as direct/indirect effectors, remain to be explored. To gain insight into the functional connection of LV in the regulation of HF behavior, we conditionally depleted LV upon the pharmacological induction of HFSC activation and during the physiological HF growth. These analyses exposed a functional connection between LV and HF growth (Fig 6).

In future studies, it will be important to analyze the contributions of LV along the different stages of the physiological HF cycle as well

as in other specific HF stages. The study of the contribution of LV at defined HF stages, along with the study of signals arising from HFSC, is particularly relevant, since the temporal diffusion of activators or inhibitors must likely lead to different responses in HF cycling in a spatiotemporal manner. Clinically, it would be relevant to expand these analyses to a human setting. Taken together, our results position LV as novel components of the HFSC niche coordinating HF connections at tissue-level and provide insight into their functional association to the HF cycle.

# Materials and Methods

## Mice and treatments

All mouse experiments were approved and performed according to international, institutional, and ethical regulations, with the approval of the local authorities.

Back skins from C57Bl/6 mice ($n$ = 3–5) from different embryonic (E15.5, E16.6, and E17.5) and postnatal (P) days (P5, P12, P16, P23, P35, P45, P49, P55, P69, and P85) were collected to analyze different phases of the HF cycle.

The Prox1-CreERT2 mouse model (Tg(Prox1-cre/ERT2)#aTmak, a kind gift from Dr. Taija Mäkinen, Uppsala University; Bazigou *et al*, 2011), was crossed under the background of the ROSA26-LSL-eYFP reporter mice. The eYFP expression was induced by intraperitoneal injections of 2 mg Tamoxifen (T5648, Sigma-Aldrich) in sunflower oil for 4 days.

Skin resident macrophages were reduced by treating P49 mice with intradermal injections of 1 mg of Clodronate encapsulated liposomes or empty liposomes as controls (Encapsula NanoSciences), as described previously (Castellana *et al*, 2014).

Cilostazol treatments were carried out by treating P49 mice with intradermal injections of 100 µg Cilostazol or vehicle control every 2 days for 9 days before sacrifice.

The Prox1-CreERT2; ROSA26-LSL-iDTR (Gt(ROSA)26Sortm1 (HBEGF)Awai/J, Jackson) mice were used to ablate LV. Mice were treated with 1 ng/g diphtheria toxin (DT) (322326, Calbiochem) via intradermal injections for 3 days. To assess LV ablation, at the end of experiments mice were intradermally injected with 10 µl of 1% Evans blue dye (E2129, Sigma-Aldrich). Skin samples were collected 16 h later and incubated in formamide (F9037, Sigma-Aldrich) at 55° for 24 h to extract the dye from the tissue. The dye levels were quantified by measuring the absorbance of the dye at 610 nm.

To analyze the effect of LV ablation upon induction of HF growth, mice were treated with the intraperitoneal administration of 100 mg/kg Cyclosporin A (CSA, 30024, Sigma-Aldrich) during 10 days starting at P49. At P52, mice were injected intraperitoneally with 2 mg Tamoxifen in sunflower oil for 4 days to induce the expression of the DT receptor (iDTR), followed by the intradermal administration of DT for 3 days starting at P56 before sacrifice at P58.

To ablate LV during the physiological HF Telogen, P49 mice were injected intraperitoneally with Tamoxifen for 4 days, followed by the intradermal administration of DT for 3 days starting at P53 before sacrifice at P55. For LV ablation during the Anagen phase of the first HF cycle, P30 mice were injected intraperitoneally with Tamoxifen for 4 days, followed by the intradermal administration of DT for 3 days starting at P34 before sacrifice at P37.

K15-CrePR1 mice [(Krt1-15-cre/PGR)22Cot/J] (Morris *et al*, 2004) and Wlstm$^{1.1Lan/J}$ mice (Carpenter *et al*, 2010) were acquired from Jackson Labs. K15-CrPR1$^{+/T}$; Wls$^{lox/lox}$ 7-day-old mice were then treated with Mifepristone (#M8046, Sigma-Aldrich) administered in the drinking water at 429 ng/ml (Babij *et al*, 2003) to the mother females and continued in weaned mice until the end of the experiment (12 weeks of treatment). K14ΔNβ-cateninER-transgenic mouse back skin samples were kindly provided by Dr. Kim Jensen (Biotech Research and Innovation Centre—BRIC, University of Copenhagen; Jensen *et al*, 2009).

## Immunofluorescence

Back skin samples were embedded in paraffin blocks and optimal cutting temperature (OCT) compound. Paraffin sections were rehydrated and treated for antigen retrieval using 10 mM sodium citrate, 0.05% Tween 20, pH 6.0 at 100°C for 10 min. Slides were incubated in blocking buffer: 0.3% Triton X-100–PBS (PBST), 5% FBS (F7524, Sigma-Aldrich), 1% gelatin from cold water fish skin (G7765, Sigma-Aldrich), and 1% BSA (10 735 078 001, Roche). Primary antibodies (Appendix Table S1) were diluted in blocking buffer and incubated overnight at 4°C and washed. The slides were then incubated for 1 h with the corresponding secondary antibodies (Appendix Table S2) and counterstained with DAPI. For OCT sections, samples were fixed in 4% PFA for 10 min and the antibody stainings were performed as described for paraffin-embedded sections.

For whole mounts and tissue clearing, embryo and postnatal back skins were fixed overnight in 4% PFA (15710, Electron Microscopy Sciences) at 4°C. The back skins were cut into small pieces, washed 10 times in PBST for 30 min each, and incubated with primary antibodies diluted in Blocking Buffer (0.3% PBST, 5% FBS and 20% DMSO) at room temperature during 5 days. Back skins were washed in PBST as previously described, followed by their incubation with secondary antibodies and counterstained with DAPI for 3 days at room temperature. Tissues were dehydrated in increasing concentrations of methanol (25, 50, and 75% methanol for 5 min) and placed in 100% methanol three times 20 min each. Next, back skins were cleared overnight in 1:2 benzyl alcohol: benzyl benzoate (BABB) (B6630 and 402834, Sigma-Aldrich).

Images were acquired in a TCS-SP5 (AOBS) confocal (Leica microsystems) microscope. High-resolution images were captured using a 63× HCX PL APO 1.3 Glycerol immersion objective.

The quantitative analyses of fluorescence intensity were performed as previously described (Verma *et al*, 2012). Briefly, regions of interest were plotted in LYVE1$^+$ cells in the vicinity of HF throughout skin sections and used to evaluate the mean fluorescence intensity of Jup and Emilin1 using ImageJ.

## Intravital confocal microscopy

Mice were anesthetized with 1.5% isoflurane and placed into a custom-made adapter with a stable temperature during the whole duration of the experiments. Intravital experiments were carried out in a TCS-SP5 (AOBS) confocal (Leica microsystems). The image acquisition was conducted during 8 h every 15 min, using a 20× HCXPL APO 0.7 N.A. dry objective.

## FACS and RNA extraction

Back skins of Prox1-CreERT2; ROSA26-LSL-eYFP mice were collected the day after the last intraperitoneal injection of 2 mg Tamoxifen. Next, the subcutaneous fat was removed with a scalpel, and the skin samples were cut into small pieces and incubated in 0.33 mg/ml of Liberase TM Research Grade (05401119001, Roche) at 37°C for 30 min. After incubation, skin samples were transferred to a MACS gentle dissociator C tube (130-093-237, Miltenyi Biotec) and filtered through 40- and 70-µm cell strainers (352350 and 352340, Falcon). Red blood cells were eliminated with lysing buffer (555899, BD Biosciences). DAPI was added as a viability dye, and FACS-isolated eYFP-positive cells (FACS ARIA IIu sorter, Becton Dickinson) were collected in Lysis Buffer (400753, Absolutely RNA Nanoprep Kit, Agilent Technologies).

For the analysis of the expression of Wls in HFSC, HFSC was FACS isolated and stained as previously described using the markers CD34 and α6 integrin (Castellana et al, 2014). RNA extraction was performed following manufacturer's instructions.

## RNA isolation and real-time PCR (RT–PCR)

1 µg of total RNA isolated from skin using TRIzol (15596026, Invitrogen) was used for cDNA synthesis using the Ready-to-Go You-Prime It First-Strand beads and random primers (GE Healthcare). RT–PCRs were conducted using the GoTaq qPCR Master Mix (A6001, Promega) and a MasterCycler Ep-Realplex thermal cycler (Eppendorf).

The expression levels were normalized to Actin. The complete list of primers used is presented in Appendix Table S3.

## RNA sequencing

Variable amounts of total RNA samples, between 50 and 2,000 pg, were processed with the SMART-Seq v4 Ultra Low Input RNA Kit (Clontech) by following manufacturer instructions. Resulting cDNA was sheared on an S220 Focused-ultrasonicator (Covaris) and subsequently processed with the "NEBNext Ultra II DNA Library Prep Kit for Illumina" (NEB #E7645). Briefly, oligo(dT)-primed reverse transcription was performed in the presence of a template-switching oligonucleotide; double-stranded cDNA was produced by limited-cycle PCR and submitted to acoustic shearing. Fragments were processed through subsequent enzymatic treatments of end-repair, dA-tailing, and ligation to Illumina adapters. Adapter-ligated libraries were completed by six cycles of PCR. Reads were sequenced in single-end mode (51 bp) on an Illumina HiSeq2500 by following manufacturer's protocols.

## Gene expression analysis by RNA-seq and Gene set enrichment analysis (GSEA)

The differential expression of genes was assessed using the Nextpresso pipeline (http://bioinfo.cnio.es/nextpresso/). The sequencing quality was analyzed with FastQC (http://www.bioinformatics.babraham.ac.uk/projects/fastqc/); reads were aligned to the mouse genome (GRCm38/mm10) using TopHat-2.0.10 (Trapnell et al, 2012), Bowtie 1.0.0 (Langmead et al, 2009), and Samtools 0.1.19.0 (Li et al, 2009). The assembly of transcripts, abundance estimation, and differential expression were calculated with Cufflinks 2.2.1 (Trapnell et al, 2012), using the mouse GRCm38/mm10 transcript annotations from https://ccb.jhu.edu/software/tophat/igenomes.shtml. We then used the RNA-seq gene list, where genes were ranked according to their statistical expression difference to perform a GSEAPreranked analysis (Subramanian et al, 2005), using pathway annotations from Reactome, BioCarta, NCI (http://www.ndexbio.org/#/user/301a91c6-a37b-11e4-bda0-000c29202374), and KEGG public databases.

All basic and advanced fields were set to default, and we considered only the gene sets with false discovery rate (FDR) $q$-values < 0.25 that were significantly enriched.

For enrichment analysis, genes with a variation of $\log_2$ fold change > 1.5 or < −1.5 were selected and introduced in the Enrichr tool (Chen et al, 2013; Kuleshov et al, 2016). The genes that belonged to the categories of interest were chosen for the heatmaps. Heatmaps representing the $\log_2(\text{FPKM} + 1)$ levels were done using the software Morpheus (https://software.broadinstitute.org/morpheus).

## In situ hybridization

Double in situ hybridization (ISH) (RNAscope 2.5 HD Duplex Assay Kit (ACD, 322440), Advanced Cell Diagnostics, Hayward, CA) was performed in P55 and P70 mouse back skin. LYVE1 was used as a lymphatic marker [Advanced Cell Diagnostics (ACD) Probe: 428451]. The expression of the following genes in $LYVE1^+$ lymphatics was analyzed as follows: ITGA5 (ACD Probe:575741), PKD1 (ACD Probe:549151), PLXND1 (ACD Probe:405931), or DCN (ACD Probe:13179).

Positive staining was indicated by red and blue punctate mRNA spots present in the nucleus and/or cytoplasm. The quantitative analyses are represented as mean of the number of mRNA spots per $LYVE1^+$ cells in the vicinity of HF throughout skin sections.

## Statistical analyses

Image analyses were performed using ImageJ and Imaris software (Bitplane Scientific Software, Zurich). For statistical analysis of quantitative data, the normality of the data was evaluated and data that presented a Gaussian distribution were analyzed using Student's $t$-test (to compare between two groups) and one-way analyses of variance (ANOVA) followed by Tukey's test (for multiple comparisons between more than two groups). Data that did not present a Gaussian distribution were analyzed using the Mann–Whitney $U$ and Kruskal–Wallis tests. Statistical analyses were done using GraphPad Software (La Jolla, CA). All data are representative of at least two independent experiments performed in triplicates.

# Data availability

RNA sequencing data have been deposited in NCBI's Gene Expression Omnibus (Edgar et al, 2002) and are accessible through GEO Series accession number GSE102463 (https://www.ncbi.nlm.nih.gov/geo/query/acc.cgi?acc=GSE102463).

**Expanded View** for this article is available online.

## Acknowledgements

We thank Dr. Kari Alitalo (University of Helsinki) for advice, Dr. Taija Mäkinen (Uppsala University) for providing the Prox1ERT2 mice, Dr. Kim Jensen (BRIC, University of Copenhagen for providing K14Cre[+/T], ΔNβ-catenin[lox/lox] mouse skin samples, and Susan Morton for the Lhx2/9 antibody (T. Jessel lab, Columbia University). We thank the core facilities of the CNIO and the University of Copenhagen for technical assistance. This work was supported by grants from the Spanish Ministry of Economy and Competitiveness/European Regional Development Fund (ERDF), European Union (BFU2015-71376-R to MP-M), the Worldwide Cancer Research UK Foundation (15-1219 to MP-M), and the Novo Nordisk Foundation (NNF17OC0028028).

## Author contributions

DP-J and SF experimental design and analyses. DM confocal and intravital microscope studies, CF-T and OG-C RNA-seq methodological analyses, DC samples and experimental analyses, and RL reagents, hypotheses discussion, and input. MP-M study conception, supervision, and funding. DP-J, SF, and MP-M article writing with input from all authors.

## Conflict of interest

The authors declare that they have no conflict of interest.

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
