## [Review Process File · The EMBO Journal]

Lymphatic vessels interact dynamically with the hair follicle stem cell niche during hair regeneration in vivo

Daniel Peña-Jimenez, Silvia Fontenete, Diego Megias, Coral Fustero-Torre, Osvaldo Graña-Castro, Donatello Castellana, Robert Loewe, Mirna Perez-Moreno

Review timeline:

Submission date:	12th Feb 2019
Editorial Decision:	12th Mar 2019
Revision received:	12th Jul 2019
Editorial Decision:	5th Aug 2019
Revision Received	5th Aug 2019
Accepted:	6th Aug 2019

Editor: Daniel Klimmeck

Transaction Report:

1st Editorial Decision

12th Mar 2019

Thank you again for your interest and the submission of your manuscript (EMBOJ-2019-101688) to The EMBO Journal. Your manuscript has been sent to three referees for consideration, and we have received reports from all of them, which I enclose below.

As you will see, the referees acknowledge the potential high interest and novelty of your work, although they also express a number of issues that will have to be addressed before they can support publication of your manuscript in The EMBO Journal. Referee #1 states that the interplay between HF cycling, differentiation and lymph vessel dynamics should be explored by additional experiments. Referee #3 argues that the analysis of the genes differentially expressed in lymph vessel during telogen phases should be expanded to consolidate the results and concept proposed. In addition, the referees point to issues related to data representation, missing controls and methods annotation that would need to be conclusively addressed to achieve the level of robustness needed for The EMBO Journal.

I judge the comments of the referees to be generally reasonable and given their overall interest, we are in principle happy to invite you to revise your manuscript experimentally to address the referees' comments.

Please note that while per se well taken, the point raised by referees #2 and #3 on conservation of the findings in human skin is not at the core of the current study in our view, thus can be left for future work.

REFeree REPORTS:

Referee #1:

Hair follicles (HFs) of the murine skin have previously been shown to be associated with and

remodel in concert with the blood vessels within the dermis. In this paper, the authors have described that lymphatic vessels (LVs), the complementary network to the cardiovascular system that circulates lymph, are also closely associated with hair follicles (HF) and may represent a niche for the stem cells of the HFs. This is a very novel idea and a valuable contribution to the field as this is a previously unexplored niche and relationship for HF regeneration. This study finds that in back skin, LVs can be found in a stereotypic organization with HFs whereby they are regularly found on the anterior side of HFs in patterns of triads. They are persistently found to be associated with the HFSC pool throughout the HF cycle and that induced growth of HFs was perturbed when LVs were ablated.

Major Concerns:

- The organization of LVs with triads of HFs is an interesting observation and suggests a tightly regulated organization that persists through HF cycling. However, the measurements of LV area in Figure EV2 and caliber in Figure 4 could be further expanded in light of the findings in Figure 5 that many pathways involving lymphatic remodeling are altered during HF growth phases. Do lymphatic endothelial cells undergo proliferation or apoptosis during phases of HF growth/regression similar to what has been observed for blood vessels?
- The finding that ablation of LVs abrogates the precocious entry of HFs into anagen is extremely compelling and suggests an important function of LVs for normal HF cycling (Figure 6). Does ablation of LVs affect the normal cycling of HFs in non-pharmacologically induced growth and regression? And furthermore does it affect the normal differentiation mechanisms that HF cells utilize during growth?

Minor Concerns:

- The abstract seems to focus on the relationship between LVs and HF stem cells but this might undercut the overall novelty of the paper as many of the results point towards LVs being important for HF development, cycling and organization as a whole, a process that could involve more than the outlined HFSCs. We suggest reframing the abstract to state the importance of LVs to the process of HF regeneration.
- The finding that specific ablation of Wntless in K15 cells results in a reorganization of associated LVs is a very interesting point, suggesting that the HFSCs could be responsible for a niche for LVs that direct their organization within the dermis (Figure 1). The authors should include in the figure legends what age these mice were.
- In Figure 2, the authors suggest that development of HFs from E17.5 onwards is coupled to recruitment of lymphatic capillaries. This is an interesting observation and makes one wonder what is the functional consequence of this early association of the HFs and LVs. We would suggest adding a section in the discussion to explore these possibilities.

Referee #2:

This manuscript carefully documents the structural and functional relationships between lymphatic vessels and hair follicles. The data are of high quality and the conclusions are well-justified and novel. The finding that depletion of lymphatic vessels can block the pharmacological induction of hair follicle growth is of particular interest. This study will likely form the foundation for future research into the role of lymphatics in the biology of the hair follicle. Specific issues that need to be addressed are as follows:

Major points

1. The manuscript documents the distribution of lymphatic vessels relative to hair follicles in the back skin of mice. It would enhance potential clinical relevance to test some human skin samples to see if this relative distribution is similar in the human setting.

Minor points

2. Figure 1B is labelled "P55" but the legend indicates the mice were P49. Were the mice in panel A P49 or P55? What does the labelling above the columns represent in panel F? What statistical test was used in F? Experimental Procedures refers to Student t test but surely multiple group testing was conducted.
3. What multiple group statistical testing was conducted for Figure 4B, Figure 6B and D and Figure

EV2B and C?

Referee #3:

Comments on "Dynamic interactions of lymphatic vessels at the hair follicle stem cell niche during hair regeneration". I read with interest this manuscript by Dr. Peña-Jimenez and colleagues. In this work, the authors investigated the distribution of lymphatic vessels (LV) in the skin and their interaction with the hair follicle (HF) unit. First, the authors studied the distribution and the dynamic modifications of lymphangiogenesis occurring during the hair cycle and then they demonstrated the importance of LV during HF regeneration. The authors nicely showed in vivo the dynamic communication through lymphatic vascularization, which potentially facilitates the spreading of signals that mediate HF cycling. The authors then performed RNAseq on FACS isolated LV cells from two stages of HF cycling to identify genes that regulated LV remodeling during HF regeneration.

This is a novel and original study that demonstrates the important role of LV in controlling HF stem cell activation and cycling. The study is well executed and I have only few comments before the publication of this study.

Major comments:

- 1/ The authors investigated the transcriptome of LV cells by comparing two telogen phases. They found potentially interesting genes but the data analysis is relatively superficial and no specific molecular candidates were selected and validated (e.g by qPCR or immunostaining) or functionally tested (e.g to check the protein expression by staining or to perform in vitro experiments).
- 2/ It could be also interesting to reinforce the relevance of these findings on human skin (e.g to check the staining of some candidate genes on human skin).

Minor comments:

- There are some typing errors that need to be corrected
Back skin instead of Backskin
- In page 10, you should complete the sentence
... where 60% of the cells represent lymphatic endothelial cells.....
- Figure 1c: The line separating epidermis and dermis should be replaced correctly.

1st Revision - authors' response

12th Jul 2019

Response to the reviewers' comments

Referee #1

1. The organization of LVs with triads of HFs is an interesting observation and suggests a tightly regulated organization that persists through HF cycling. However, the measurements of LV area in Figure EV2 and caliber in Figure 4 could be further expanded in light of the findings in Figure 5 that many pathways involving lymphatic remodeling are altered during HF growth phases. Do lymphatic endothelial cells undergo proliferation or apoptosis during phases of HF growth/regression similar to what has been observed for blood vessels?

In the new Fig panels EV2D and EV2E, we now show the quantification of the number of proliferating lymphatic cells (BrdU⁺, LYVE1⁺) and apoptotic lymphatic cells (cleaved caspase 3⁺, LYVE1⁺) using double immunofluorescence analyses during different HF cycle stages (P5, P12, P16, P23, P35, P45, P49, P55, P70, P85). Except for the HF stages P5, P12, and P16, no changes in LV proliferation were observed (Fig EV2D), suggesting that LV were still growing and reorganizing to HF growing from morphogenesis. Also, no changes in LV cell death were observed during the HF cycle (Fig EV2E). These new data revealed that lymphatic cells do not undergo proliferation or

apoptosis during HF growth/regression phases.

2. The finding that ablation of LVs abrogates the precocious entry of HFs into anagen is extremely compelling and suggests an important function of LVs for normal HF cycling (Figure 6). Does ablation of LVs affect the normal cycling of HFs in non-pharmacologically induced growth and regression? And furthermore does it affect the normal differentiation mechanisms that HF cells utilize during growth?

We thank the reviewer for pointing out the relevance of addressing the functional connection between LV and normal HF cycling in non-pharmacologically induced conditions.

As the referee nicely suggested, we now include the new Fig panels 6G-J and new Fig EV5C-F. The data in these figures include the effect of depleting LV in mouse skin at the Anagen phase of the first HF cycle, and analyses of the presence of LV, HF apoptosis, and the expression of differentiation markers.

These new results show that the ablation of LV prompts to a collapse of growing HF (new Fig 6G and H), accompanied with cell death (Fig 6I and J) and loss of the growing HF differentiated layers (new Fig EV5C-F). Overall, the data support a role for LV in sustaining the proliferation of HF, in agreement with the findings observed during the pharmacological induction of HF growth.

3. Minor concerns: The abstract seems to focus on the relationship between LVs and HF stem cells, but this might undercut the overall novelty of the paper as many of the results point towards LVs being important for HF development, cycling and organization as a whole, a process that could involve more than the outlined HFSCs. We suggest reframing the abstract to state the importance of LVs to the process of HF regeneration.

We agree with the referee and modified the abstract accordingly.

4. Minor concerns: The finding that specific ablation of Wntless in K15 cells results in a reorganization of associated LVs is a very interesting point, suggesting that the HFSCs could be responsible for a niche for LVs that direct their organization within the dermis (Figure 1). The authors should include in the figure legends what age these mice were.

We have included the missing information in the Figure legend 1G.

5. Minor concerns: In Figure 2, the authors suggest that development of HFs from E17.5 onwards is coupled to recruitment of lymphatic capillaries. This is an interesting observation and makes one wonder what is the functional consequence of this early association of the HFs and LVs. We would suggest adding a section in the discussion to explore these possibilities.

We agree with the referee's suggestion and added a section in the discussion related to this important aspect.

Referee #2

1. The manuscript documents the distribution of lymphatic vessels relative to hair follicles in the back skin of mice. It would enhance potential clinical relevance to test some human skin samples to see if this relative distribution is similar in the human setting.

We thank this reviewer and reviewer 3 for making this point. Analyzing the distribution and functional implications of lymphatic vessels in human skin will enhance the potential clinical relevance of our findings. We mention this aspect in the discussion section; however, given that LV exhibit a differential distribution according to their anatomical location, we would like to document any future human studies as a complete study on its own.

2. Minor points: Figure 1B is labelled “P55” but the legend indicates the mice were P49. Were the mice in panel A P49 or P55? What does the labelling above the columns represent in panel F? What statistical test was used in F? Experimental Procedures refers to Student t test but surely multiple group testing was conducted.

Those mistakes are now corrected in the figure legends of Fig 1B and 1F, and the experimental procedures' section.

3. Minor points: What multiple group statistical testing was conducted for Figure 4B, Figure 6B and D and Figure EV2B and C?

We apologize for this oversight. We indeed conducted multiple comparisons between groups, using the statistical one-way Analyses of Variance (ANOVA), with Tukey's post hoc tests. The Tukey's multiple comparison test was selected over the Dunnett's test, as Tukey's determines which means amongst a set of means differ from the rest, while the Dunnett's test compares each sample with a single control.

This information has been indicated in all of the figure legends and the experimental procedures' section.

Referee #3

1. The authors investigated the transcriptome of LV cells by comparing two telogen phases. They found potentially interesting genes but the data analysis is relatively superficial and no specific molecular candidates were selected and validated (e.g by qPCR or immunostaining) or functionally tested (e.g to check the protein expression by staining or to perform in vitro experiments).

We thank the reviewer for raising this critical point, which has helped us to substantiate our findings.

To validate the expression changes of relevant candidates in tissue, double in situ hybridization analyses (RNAscope, Advanced Cell Diagnostics, USA) for LYVE1 and selected candidates were conducted in P55 and P70 mouse skin samples. Also, we performed immunofluorescence analyses of two membrane proteins.

We now include the quantification of those findings and representative images in the new Fig panels 5D-I, and the new Fig EV3. We have also described in detail in the methods section, how these analyses were conducted, and the quantification procedure.

2. It could be also interesting to reinforce the relevance of these findings on human skin (e.g to check the staining of some candidate genes on human skin).

We thank this reviewer and reviewer 2 for making this point. See answer to reviewer 2, point 1.

3. Minor comments: There are some typing errors that need to be corrected: Back skin instead of Backskin. In page 10, you should complete the sentence ... where 60% of the cells represent lymphatic endothelial cells..... Figure 1c: The line separating epidermis and dermis should be replaced correctly.

We corrected those mistakes and revised all the text and figures.

2nd Editorial Decision

5th Aug 2019

Thank you for submitting your revised manuscript for consideration by The EMBO Journal. Your amended study was sent back to two of the original referees for re-evaluation, and we have received comments from both of them, which I enclose below. As you will see the referees find that their concerns have been sufficiently addressed and they are now broadly in favour of publication.

Thus, we are pleased to inform you that your manuscript has been accepted in principle for publication in The EMBO Journal, pending some minor issues related to formatting and data representation, which need to be adjusted at re-submission.

REFeree REPORTS:

Referee #1:

This revised manuscript has incorporated all requested edits and includes additional experiments that have provided an important distinction between the relationship of lymphatic vessels compared to blood vessels with regards to hair follicle cycling. Additional experiments involving ablation of lymphatic vessels further corroborate the importance of these vessels for normal entry into growth phase. We recommend no further experiments or edits as this manuscript is ready for publication and will make an important contribution to the field.

Referee #3:

The authors adequately addressed my initial questions and I recommend publication of this paper in EMBOJ.

2nd Revision - authors' response

5th Aug 2019

The authors performed the requested editorial changes.

Corresponding Author Name: Mirna Perez-Moreno

Manuscript Number: EMBOJ 2019-101688R